# Deep GNN-driven Surrogate for the Better Meta-learning in AutoML

**Nikolay Nikitin, Konstantin Cherniak, Mardyshkin Rostislav, Egor Shikov, Peter Shevchenko**
**Illarion Iov, Maiia Pinchuk, Grigorii Kirgizov, Andrey Stebenkov, Anna Kalyuzhnaya**
NSS Lab
ITMO University
Saint Petersburg, Russia
`{illariov1809,nicl.nno}@gmail.com`

## Abstract

Automated Machine Learning (AutoML) aims to accelerate the process of solving machine learnging (ML) problems by providing tools for automating pipeline design. However, existing AutoML approaches are often computationally expensive, as they solve high-dimensional optimization tasks without leveraging knowledge from past solutions. Meta-learning aims to leverage past experience to improve the efficiency of solving new machine learning problems. In the context of automated pipeline design, meta-learning can facilitate the ranking of candidate pipelines by drawing structural insights from a database of previously solved tasks. However, existing meta-learning approaches tend to focus on relatively simple pipelines and tasks.

In this paper, we propose using Graph Neural Networks (GNNs) as probabilistic ranking surrogates for evolutionary optimization of pipelines with variable structures in AutoML. The GNNs are trained on meta-knowledge from a database of tabular classification problems to efficiently rank candidate pipelines based on their expected performance. This enables stronger initial estimates for optimization and accelerates convergence by leveraging surrogate evaluation of the fitness function. Our approach is implemented as an open-source library that can enhance the performance of state-of-the-art AutoML solutions.

## 1 Introduction

Automated Machine Learning (AutoML) fills the important gap between ML algorithms and their practical application in science and industry Singh & Joshi (2022); Baratchi et al. (2024). There are many AutoML solutions that can design the pipeline structure using different optimization techniques He et al. (2021). The structure of the pipeline can be described by different patterns – from linear sequence to directed acyclic graph (DAG) with variable structure Zöller & Huber (2021).

However, most existing AutoML solutions are computationally expensive due to the underlying optimization approach, which requires evaluating a large number of ML pipelines. As a result, it can be difficult to apply AutoML to large datasets. The high consumption of computational resources also raises sustainability concerns Tornede et al. (2021).

Meta-optimization techniques can be used to produce solutions to AutoML tasks in a more effective way Ye & Ye (2022). Accumulated knowledge of dataset-pipeline matches can facilitate the disrovery of new solutions with minimal effort. The key challenge is to match new tasks with previously solved ones that have a considered optimal solution. This challenge can be addressed by leveraging the meta-features of each dataset.

In this paper, we propose an approach based on a deep neural graph model to improve the efficiency of evolutionary AutoML. This is achieved by replacing some of the function evaluation operations with probabilistic pairwise comparison of individuals. The approach is implemented as a modular open source framework and can be used as part of existing state-of-the-art AutoML tools.

## 2 RELATED WORKS

### 2.1 AUTOML FRAMEWORKS

The AutoML field is already well-known to a public and has several excellent surveys Baratchi et al. (2024); Barbudo et al. (2023). This section is focused on providing a brief overview of the state-of-the-art AutoML solutions.

The auto-sklearn Feurer et al. (2015) is the most popular Meta-AutoML tool. Basically, it is a wrapper for the scikit-learn Pedregosa et al. (2011) package, which additionally allows you to automate the algorithm selection and hyperparameter tuning. The underlying optimization mechanism is a Bayesian optimization (from the SMAC library).

Modern AutoML tools are focused on creating multi-layered ensembles of pipelines. For example, TPOT Olson & Moore (2016) uses genetic algorithms that consider machine learning models as primitives, and these are combined into a tree-based pipeline. AutoGluon Erickson et al. (2020) uses multi-layered stack ensembling with repeated k-fold bagging. Another approach is H2O AutoML LeDell & Poirier (2020). The key aspects of this approach are the ability to handle missing and categorical data natively, a comprehensive modeling strategy, and use of stacked ensembles. Another evolutionary AutoML approach is implemented in FEDOT Nikitin et al. (2022). It is based on the idea of evolutionary design of graph-based pipelines. A recent approach is LightAutoML Vakhrushev et al. (2021), which applies two-layer ensemble pipelines.

### 2.2 ADVANCED OPTIMIZATION TECHNIQUES IN AUTOML

Over the past few years, Reinforcement Learning (RL) techniques have been used to develop various approaches in AutoML methods. Inspired by the success of AlphaeZero Silver et al. (2017), Drori's research group suggested AlphaD3M Drori et al. (2021). AlphaD3M is an automatic machine learning system that uses DARPA's Data-Driven Discovery of Models (D3M) as a foundation. The core idea of AlphaD3M is to use the neural network for predicting pipeline performance and action probabilities along with a Monte-Carlo Tree Search that takes strong decisions based on the network. The regularization problem is also considered as part of the pipeline design task Nikitin et al. (2024).

Most of the AutoML methods discussed start by building a pipeline for a new, unknown ML task from scratch. However, existing AutoML approaches are generally computationally expensive because they solve high-dimensional optimization tasks without using knowledge from previously solved tasks.

### 2.3 META-LEARNING

One possible direction for improving AutoML is the meta-learning approach. It extracts knowledge from previous runs of AutoML and the known performance of ML pipelines on ML tasks to improve AutoML. For example, the choice of search space directly affects the results of AutoML Cambronero et al. (2020). In Rakotoarison et al. (2021), appropriate reductions in the search space can be achieved through meta-learning that takes into account the experience of previous runs. Different patterns can be extracted from the final structures Zöller et al. (2021) obtained in an automated way. Meta-learning can use meta-characteristics of datasets to suggest relevant pipelines as an initial search point (warm-starting). Meta-learning can use a pairwise correlation matrix of ML models describing their co-occurrence in pipelines. The structure of pipelines or features of datasets can also be represented as an embedding Fusi et al. (2018); Singh et al. (2021); Jomaa et al. (2021).

In evolution-based AutoML, the parameters of evolutionary algorithms (EA) can be adjusted dynamically during optimization, as different parameter values may be optimal at different stages of the optimization process and for different tasks Aleti & Moser (2016). An overview of adaptive and machine learning approaches in EA is given in Aleti & Moser (2016) and Mamaghan et al. (2021). For example, the frequency of mutations can vary depending on the diversity of the current population Evans et al. (2020). In particular, the subfield of Adaptive Operator Selection (AOS) adapts the probabilities of evolutionary operators using optimization history Fialho & Roberto (2010). In AOS, there are approaches using Multi-Armed Bandits (MAB) Li et al. (2014) and attempts to use

Reinforcement Learning (RL) Drugan (2019); Buzdalova et al. (2014). However, such approaches only take into account the experience gained during the current run or use some static heuristics.

Overall, we can highlight three approaches to control the optimization process through the meta-algorithm: (1) changing initial assumptions and parameters; (2) restricting the search space; (3) and guided exploration of the search space.

The latest meta-AutoML approaches try to use large language models (LLMs) Xu et al. (2024); Zhang et al. (2023). This allows experiments to be designed using descriptions of data and models and to be carried out in an automated way for various tasks (including computer vision and natural language processing). The hybridization of GPT-like LLMs and evolutionary algorithms for NAS tasks can also be used to improve the efficiency of AutoML and reduce optimization time Yu et al. (2023). There is also an example of a successful application of the GPT model to automate feature engineering Hollmann et al. (2023). The authors of the survey Gu et al. (2024) on the use of LLMs for AutoML applications provide a comprehensive review of the application of LLMs in AutoML, specifically in data and feature engineering, model selection, hyperparameter optimization, and workflow optimization. While this approach demonstrates promising results, several significant challenges remain, including the risk of data leakage, which can compromise result validity, the complexity of prompt engineering, the occurrence of hallucinations, and difficulties in interpretability. A key limitation of employing LLMs in AutoML is their substantial resource consumption, which exceeds that of alternative methods. Even when LLMs yield superior results, their high computational cost may render them impractical in certain scenarios.

These examples confirm the great potential of different variants of meta-optimization for improving AutoML approaches. However, there is no ready-made solution for the automated design of complex graph-based modeling pipelines that can transfer knowledge between similar tasks and provide an always-on solution for AutoML tasks.

## 2.4 Pipeline Ranging

To train a ranking model for machine learning pipelines, it is necessary to represent pipelines as embeddings. One way to represent pipeline is a heterogeneous graph, where nodes correspond to machine learning algorithms and their associated hyperparameters.

Several approaches can be employed to transform a heterogeneous graph into an embedding, including random walk-based methods, graph neural networks (GNNs), and deep learning approaches that do not rely on GNNs. The work of Dong et al. (2017) extends the conventional random walk-based method, Path2Vec (Kutuzov et al. (2018)) by incorporating meta-path guided random walks to better capture relationships between nodes. Among the various approaches, GNN-based methods are the most widely adopted, particularly their heterogeneous graph extension, relational graph convolutional networks (RGCN) Schlichtkrull et al. (2017). Alternative methods employ different node aggregation mechanisms, such as the Transformer-based architecture utilized in Heterogeneous Graph Transformer Hu et al. (2020). More recent techniques, including Graph-BERT Zhang et al. (2020), deviate from conventional message-passing framework of GNNs. Instead, they leverage Transformer architectures to explicitly model the structural dependencies between nodes.

Once the pipeline is represented as a graph embedding, the next step is to rank pipelines based on their suitability for a given problem. The most commonly used learning-to-rank approaches fall into three categories: pointwise, which treats ranking as a regression or classification problem for individual pipelines; pairwise, which optimizes the relative ordering between two pipelines; and listwise, which directly models ranking functions based on entire sets of candidate pipelines Liu et al. (2009).

## 3 Problem statement

The problem statement of automated machine learning can be considered as a discrete optimization task. The following formulation can be used:

$$P^* = \underset{P \in \mathbb{P}}{\arg\max} \, f\left(P | T_{gen} \leq \tau_g\right),$$

$$P = \langle G, E \rangle, G = A(\mathbf{D}, \{\lambda\}_n), P \in \mathbb{P}, \tag{1}$$

where $f$ is the black-box objective function that characterizes the efficiency of the pipeline $P$ for dataset $\mathbf{D}$ and $P^*$ is the best obtained pipeline using a discrete optimization algorithm, $\mathbb{P}$ is a pipeline search space, $T_{gen}$ is the time spent on optimization, $\tau_g$ is the time limit. A pipeline $P$ itself is formulated as a DAG with nodes $G$. Each node $G$ contains a machine learning algorithm $A$ that trains on data $\mathbf{D}$ with hyperparameters $\lambda$.

The main problem is that $\tau_g$ should be large enough to achieve the appropriate number of iterations of the optimization algorithm. At the same time, each evaluation of the fitness function can be very computationally expensive due to the complexity of the pipeline and the size of the training sample.

To reduce $T_{gen}$ without reducing the quality of the obtained pipelines, we can move from Eq. 1 to a meta-optimization formulation, which uses information extracted from historical data $\mathbf{D}_{prev}$ about previous learning results to make the time spent on meta-optimization $T_{meta}$ much smaller than the time spent on optimization $T_{gen}$:

$$P^* = \underset{P \in \mathbb{P}}{\arg\max} \, f\left(P | T_{meta} \leq \tau_g, \mathbf{D}_{prev}\right), \;\; T_{meta} << T_{gen}, \tag{2}$$

There are several meta-optimization strategies that can be applied. We can select $P_{best}$ — the existing pipeline for the most similar datasets from $\mathbf{D}_{prev}$. We can also train a generative model using $D_{prev}$ as a training sample. Finally, we can use the knowledge extracted from $T_{gen}$ to improve the convergence of the optimization algorithm (e.g. evolutionary search) at different stages: choice of the initial assumption, evaluation of the objective function, and fine-tuning of the algorithm hyperparameters for $\mathbf{D}_{prev}$.

In the case of objective-function approximation, a Graph NN (GNN) can be used in Eq. 3. For GNN model training, information about previous attempts to solve similar tasks from sets $\boldsymbol{D_{prev}}$, machine learning algorithms trained on data $\boldsymbol{D_{prev}}$, and resulting numerical values of the obtained quality metric is used. After GNN training, it can be used as a surrogate model for new data $\mathbf{D}_{new}$.

$$\hat{f} = GNN(\mathbf{D}, A(\mathbf{D}, \{\lambda\}_n)),$$

$$\mathbf{D} = \{\mathbf{D}_{prev}, \mathbf{D}_{new}\}, \tag{3}$$

However, we do not need to evaluate the exact values of the objective function. If we use the model to compare the values of the function, the GNN-based surrogate can be considered as an order-oracle Lobanov et al. (2024) for the optimization.

## 4 PROPOSED APPROACH FOR SURROGATE MODELING OF PIPELINE OBJECTIVE WITH GNN

We design a modular and effective GNN-based solution to the problem of Meta-AutoML. It is based on a multi-stage approach: (1) the construction of a meta-knowledge base (meta-storage); the (2) fast selection of initial assumptions using GNN; (3) evolutionary optimization with surrogate estimation of the objective function for a pipeline.

The surrogate model is used for the fast estimation of the objective function. Pipeline-fitness evaluation is one of the most expensive operations in AutoML algorithms. A database of previous runs of AutoML on various ML datasets $\mathbf{D}_{prev}$ contains a large number of pipeline-score pairs. It allows for training a surrogate model for estimating pipeline fitness and partially bypassing expensive pipeline evaluation on real data during evolutionary optimization. The primary part with sensors is described in Figure 1.

The general surrogate model scheme is as follows: we form embeddings of the dataset and the pipelines to be compared, and we train the surrogate model via backpropagation to rank the pipelines

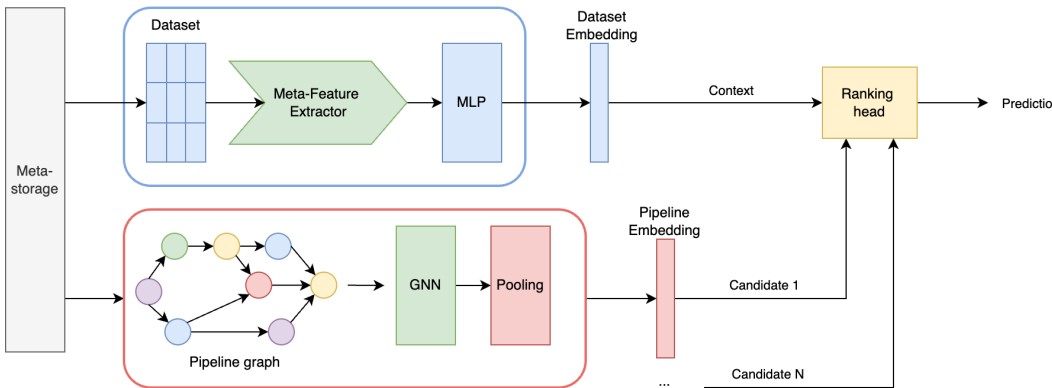

Figure 1: Architecture of GNN-based oracle proposed as a part of an evolutionary optimizer. Dataset and pipelines are mapped to their embeddings and then passed to the ranking head for target prediction.

using the dataset embedding as a query. In this work, we define the correct ordering as pipelines with a better quality metric having a higher rank.

Our choice of a ranking task rather than direct metric prediction is due to the fact that the same composite ML models can be applied to different ML tasks, and different metrics can be used within the same task. Therefore, it is useful to predict the ranking of models in terms of their quality on a given dataset.

In this work, we have explored three different ranking options: pointwise, pairwise and listwise; and for the pairwise and listwise options, we have explored early and late stages for introducing a query.

**GNN pipeline encoder.** A pipeline $P$ characterized by different algorithms $A$, each defined by its type $T_A$ and associated hyperparameters $\lambda_A$, forms a heterogeneous graph $G_{het}$. This heterogeneity arises from the varying nature of the algorithms, each accompanied by its unique set of hyperparameters $\lambda_A$, which differ both in size and semantics. However, despite this variance, the general type of each node maintains a consistent general type — they all represent algorithms, ensuring uniformity in the edges between them.

Given the differing semantics of these features, a simplistic approach such as filling the hyperparameters with zeros to homogenise the graph is not viable. Therefore, to facilitate the use of a homogeneous GNN for embedding such graphs, we propose using specialised encoders $\text{Enc}_A$ per algorithm. These encoders are implemented as Multi-Layer Perceptrons (MLPs). They serve to translate the hyperparameters of each operation into a shared space $\lambda \leftarrow \text{Enc}_A(\lambda_A)$, thereby achieving graph homogeneity. For algorithms with hyperparameters, encoders consist of multiple feed-forward layers, while for algorithms without hyperparameters, encoders are learnable vectors.

The algorithm name is processed via ordinal encoding and mapped to a learnable vector using an embedding layer, which is then concatenated to the homogenised hyperparameter feature $\lambda$. This homogeneous graph is then processed using a GNN to produce the graph embedding.

**Dataset encoder.** The encoder uses a meta-features extractor to obtain a representation of the data, which is then normalized and passed through MLP, denoted as $\text{MLP}_{dset}$. In this paper, we use the metafeatures from PyMFE and OpenML listed in Alcobaça et al. (2020)) and Bilalli et al. (2017).

**Dataset and pipeline encoders** are jointly trained with a ranking head in an end-to-end fashion.

**Ranking head.** This approach Burges et al. (2005) transforms the surrogate into a scoring function with a $\text{MLP}_{score}$ layer that accepts a concatenation of a meta-feature embedding and a pipeline embedding to predict the pipeline score for the given dataset. The final ranking is achieved by using the scores to sort a candidate set.

Given a meta-feature vector $d$ and pipelines $P_A$ and $P_B$ with metrics $s_A$ and $s_B$, training the surrogate model $f_\omega$ parameterized with weight $\omega \in \Omega$ is as follows:

$$
\begin{aligned}
\omega &\leftarrow \underset{\omega \in \Omega}{\operatorname{argmin}} \operatorname{BCELoss}\left(y_{AB}, \rho_{AB}\right), \\
\rho_{AB} &\leftarrow \operatorname{Sigmoid}\left(O_{AB}\right), \\
O_{AB} &\leftarrow Z_A - Z_B, \\
Z_i &\leftarrow \underset{\omega \in \Omega}{\operatorname{MLP}_{score}}\left(E_i\right), \\
E_i &\leftarrow \operatorname{concatenate}\left(x_i, x_d\right), \\
x_i &\leftarrow \underset{\omega \in \Omega}{\operatorname{GNN}}\left(P_i\right), \\
x_d &\leftarrow \underset{\omega \in \Omega}{\operatorname{MLP}_{dset}}\left(d\right), \\
y_{AB} &\leftarrow
\begin{cases}
1 & \text{if } s_A > s_B \\
0.5 & \text{if } s_A = s_B \\
0 & \text{otherwise}
\end{cases}
\end{aligned}
\tag{4}
$$

During inference, $Z_i$ values are used as pipeline scores. These values are a uniform representation of a pipeline metric, allowing the surrogate to be trained on different metrics, provided that the metric is the same for a pair of pipelines. For a schematic overview of the proposed methods, refer to Appendix G.

**DirectRanker head.** This approach transforms the surrogate into a comparison function with a $\operatorname{MLP}_{comp}$ layer to predict whether the first pipeline is better than the second for the given dataset. The final ranking is achieved by using the comparator in any sorting algorithm.

This method is adopted from Köppel et al. (2020) as it claims all the required properties of a correct comparator. A similar idea of combining GNN and DirectRanker can be found in Damke & Hüllermeier (2021). However, we apply a different strategy to graph embedding and introduce a query for ranking.

For this method, we have studied two stages of introducing a query to the model.

For the early fusion, the training of the surrogate model is the same as in Eq. 4 with the following changes:

$$
\begin{aligned}
\omega &\leftarrow \underset{\omega \in \Omega}{\operatorname{argmin}} \operatorname{MSE}\left(y_{AB}, \rho_{AB}\right), \\
\rho_{AB} &\leftarrow \operatorname{Tanh}\left(O_{AB}\right) \\
O_{AB} &\leftarrow \underset{\omega \in \Omega}{\operatorname{MLP}_{comp}}\left(Z\right), \\
Z &\leftarrow E_A - E_B, \\
E_i &\leftarrow \underset{\omega \in \Omega}{\operatorname{MLP}_{join}}\left(J_i\right), \\
J_i &\leftarrow \operatorname{concatenate}\left(x_i, x_d\right), \\
y_{AB} &\leftarrow
\begin{cases}
1 & \text{if } s_A > s_B \\
0 & \text{if } s_A = s_B \\
-1 & \text{otherwise}
\end{cases}
\end{aligned}
\tag{5}
$$

For late fusion, the surrogate model training is the same as in Eq. 5 with the following changes:

$$
\begin{aligned}
Z &\leftarrow \operatorname{concatenate}\left(x_p, x_d\right), \\
x_p &\leftarrow x_A - x_B
\end{aligned}
\tag{6}
$$

During inference, $\rho_{AB}$ is rounded to an integer and used as the result of comparing two pipelines $P_A$ and $P_B$. This allows the surrogate to be trained on different metrics provided that the metric is the same for a pair of pipelines.

**SetRank head.** This approach Pang et al. (2020) allows the surrogate to accept an arbitrary number of pipelines and a meta-feature vector to evaluate the pipelines against each other. The variable

size of a set of pipelines is achieved with the Self-Attention Transformer $\underset{\omega \in \Omega}{\text{SAT}}$, which can handle sequences of arbitrary length. No positional encoding is required here, since a set of candidates to be ranked is an unordered sequence. The final ranking is achieved by using the scores to sort a set of candidates.

For this method, we considered two stages of introducing a query into the model. We denote a set of $N$ pipelines as $P \leftarrow \{P_1, P_2, ..., P_N\}$ and their corresponding metrics as $S \leftarrow \{s_1, s_2, ..., s_N\}$.

For early fusion, the training of the surrogate model is the same as in Eq. 4 with the following changes:

$$
\begin{aligned}
\omega &\leftarrow \underset{\omega \in \Omega}{\arg\min} \text{CrossEntropyLoss}\left(y, \rho\right), \\
\rho &\leftarrow \text{Softmax}\left(O\right), \\
O &\leftarrow \underset{\omega \in \Omega}{\text{MLP}_{score}}\left(Z\right), \\
Z &\leftarrow \underset{\omega \in \Omega}{\text{SAT}}\left(E\right), \\
E &\leftarrow \text{concatenate}\left(E_1, E_2, ..., E_N\right), \\
y &\leftarrow \text{Softmax}\left(S\right)
\end{aligned}
\tag{7}
$$

For late fusion, a cross-attention transformer layer $\underset{\omega \in \Omega}{\text{CAT}}$ is introduced to reweight the pipeline scores $U$ with respect to the given dataset $x_d$. The training of the surrogate model is the same as in eq. 7 with the following changes:

$$
\begin{aligned}
Z &\leftarrow \underset{\omega \in \Omega}{\text{CAT}}\left(U, x_d\right), \\
U &\leftarrow \underset{\omega \in \Omega}{\text{SAT}}\left(X\right), \\
X &\leftarrow \text{concatenate}\left(x_1, x_2, ..., x_N\right)
\end{aligned}
\tag{8}
$$

During inference, $\rho$ values are used as pipeline scores. These values represent the position of pipelines in a set. This allows the surrogate to be trained on different metrics, since the metric is the same for each set of pipelines.

The proposed meta-learning framework that implements the GNN-based optimization has a modular architecture of components to make it adaptable to different AutoML setups.

## 5 EXPERIMENTAL STUDIES

### 5.1 EXPERIMENTAL SETUP

The setup includes several configurations of AutoML systems. The baseline part includes the AutoML framework FEDOT and the AutoSklearn 2.0 framework (as an example of an existing meta-AutoML solution). The results can also be compared with state-of-the-art solutions using metrics from the widely used AutoML benchmark Gijsbers et al. (2024). For the experiments, we configured a server based on Xeon Cascadelake (2900MHz) with 8 cores and 48GB memory.

The experiments were performed on data prepared using the self-developed AutoML framework. The data includes 11 classification datasets from the OpenML platform, each forming 10 cross-validation folds. The total number of pipelines collected is 2,499,548. For each metadata set, 189 to 1,869 unique architectures of pipelines are generated. For each unique architecture, 50 variants with different hyperparameters are provided. The dataset is divided into training and validation sets as 80% of pipelines and 20% of pipelines respectively.

### 5.2 HYPERPARAMETERS

Here we provide some additional details related to hyperparameters of the model.

**Algorithm Encoder.** In this paper we translate the hyperparameters of any algorithm into a shared space with a MLP of 2 layers and a hidden size of 8 at all layers. To translate the algorithm name, we encode it with Ordinal Encoding and map it to a learnable vector of length 2. For algorithms without hyperparameters, we use a learnable vector of the same dimension as the output of the above MLP. After concatenating the features, the resulting node size is 10.

**Graph Encoder Parameters.** A 2-layer graph convolutional network with a hidden size of 64 was used for pipeline embedding. We set the dropout to 0.3 and used the BatchNorm layer. The final pipeline representation was obtained using mean pooling.

**Dataset Coder Parameters.** A 2-layer MLP with hidden size 16 and ReLU nonlinearity was used for the dataset encoding.

**Rank head parameters.** The ranking head is a 2-layer MLP with hidden size 16 and ReLu non-linearity. Early fusion DirectRanker and late fusion DirectRanker are a 1-layer MLP with hidden size 1 with early fusion option using additional 2-layer MLP with hidden size of 16 and ReLu nonlinearity to join a pipeline and dataset embeddings. Early fusion SetRank head is a 4-layer Self-Attention Transformer with 8 heads and 0.1 dropout. Late fusion SetRank head is the same with the addition of a 2-layer Cross-Attention Transformer with 8 heads and 0.1 dropout.

**Optimizer and loader parameters.** A dataloader with a batch size of 256 candidate dataset combinations was used for training. Each candidate set is randomly formed from 10 pipelines with the same metric name and the same accompanying dataset. We did this in order to reuse the same training dataset class. For point-wise and pair-wise methods, we get 9 pairs from the set for free, as the set is sorted based on the true scores of the pipelines in descending order. This gives us a batch size of 1152 for both point-wise and pair-wise methods. We used an Adam optimizer with a learning rate of 0.001 and a weight decay of 1e-4.

### 5.2.1 SURROGATE COMPARISON

To evaluate the developed surrogate models, we use the Kendall-Tau correlation coefficient and the precision score. Although NDCG score is a common metric to assess the quality of a ranking model, this metric turns out to be unsuitable for the data used in this work because the relevance scores are close to each other (e.g., 0.9 and 0.91), which makes the NDCG score insensitive to permutations of the ranked candidate set.

To train the SetRank approach, candidate sets are formed with 10 pipelines, these quality metric values are transformed into relevance by applying a softmax function over a set of candidates.

For the evaluation of all methods, candidate sets are generated in the same way as for the SetRank approach. During the evaluation, 5000 candidate sets are randomly selected across all metadata sets.

The results show that the RankNet head produces the best result regardless of the meta-features used (Table 1). A comparison of the early and late fusion strategies shows that the early fusion strategy produces higher results. Furthermore, the particular set of meta-features used does not lead to significant differences in the results. Figures Figures 2 and 3 in the Appendix show plots of mean predicted relevance versus true relevance with standard deviation. On average, the surrogates predict correctly, although the deviation is quite significant.

### 5.2.2 FEATURE IMPORTANCE

This section presents an ablation study to investigate whether the surrogate effectively utilises all the features provided.

The surrogate model accepts a pipeline represented as a heterogeneous graph and metadata features as a vector. The graph is characterised by edges between nodes (operations), and each node is characterised by a type and hyperparameters, resulting in three input features. The metadata vector is considered as the fourth input feature.

To evaluate the importance of each feature, we conducted experiments where each feature was shuffled independently; the resulting change in surrogate quality was measured. The experiment is provided on PyMFE meta-features with 5000 candidate sets over all the meta-datasets as in the evaluation section.

Table 1:   Comparison of different ranking head setups for proposed architecture of surrogate model.

| Surrogate | Fusion | Kendall-Tau | Precision |
|---|---|---|---|
| OpenML Meta-features | | | |
| Random | - | 0.00 | 0.10 |
| Proposed RankNet-based | Early | **0.52** | **0.24** |
| Direct Ranker | Early | 0.51 | 0.24 |
| | Late | 0.42 | 0.19 |
| SetRank | Early | 0.44 | 0.21 |
| | Late | 0.41 | 0.2 |
| PyMFE Meta-features | | | |
| Random | - | 0.00 | 0.10 |
| Proposed RankNet-based | Early | **0.53** | **0.25** |
| Direct Ranker | Early | 0.52 | 0.24 |
| | Late | 0.41 | 0.19 |
| SetRank | Early | 0.44 | 0.21 |
| | Late | 0.36 | 0.18 |

Table 1 summarizes the drop in surrogate quality for different permutations. For DirectRanker and SetRank both the early and late fusion cases are presented, while for RankNet, only a single embedding is used to assess the pipeline, so no fusion options are available. The results show that all provided features are important, although their importance varies. Graph-related features are more important than meta-dataset features. This fact is consistent with the initial assumption that metadata features only serve as a query that tells the surrogate how to rank pipelines. Interestingly, late-fusion strategies show the least use of metadata information. This is a possible reason why the late-fusion surrogates show the worst quality. We assume that there is no key insight in the fact that late-fusion surrogates rely mostly on connections between nodes, as the importance of this feature is approximately the same as for the other surrogates.

Table 2:    Surrogates features importance.

| Surrogate | Fusion | Op. type, %($\downarrow$) | Op. hparams, % | Graph edges, %($\downarrow$) | Dataset meta-feats, % |
|---|---|---|---|---|---|
| Kendall-Tau | | | | | |
| RankNet | Early | **-77** | -40 | -62 | -38 |
| Direct Ranker | Early | **-80** | -41 | -63 | -37 |
| | Late | -50 | -46 | **-68** | -24 |
| SetRank | Early | **-66** | -36 | -57 | -36 |
| | Late | -58 | -42 | **-58** | -19 |
| Precision | | | | | |
| RankNet | Early | **-50** | -29 | -42 | -29 |
| Direct Ranker | Early | **-50** | -29 | -46 | -29 |
| | Late | -35 | -30 | **-40** | -5 |
| SetRank | Early | **-38** | -24 | -33 | -24 |
| | Late | -28 | -22 | **-33** | -6 |

### 5.2.3   PIPELINE GENERATION

In this section, the proposed surrogate with RankNet head is integrated into the FEDOT framework to investigate its impact on both the generation process duration and the resulting pipeline quality. We then compare the obtained results with the default baseline configuration and proposed surrogate-based setup.For pipelines taken from the database, we assume that the pipeline generation time $T$ is zero.

For FEDOT we restrict possible algorithms to the set of algorithms in the training database, so that the surrogate has a corresponding encoder for an algorithm. LGBM is used as initial assumption.

For this experiment, we run AutoML on OpenML datasets that are not represented in test. The experiment was performed using OpenML features due to their ease of acquisition.

Table 3 shows the results of the comparison of criteria between surrogate and non-surrogate setups. For our framework, the surrogate allows to drastically reduce the time needed to generate a pipeline with insignificant quality reduction while maintaining similar quality (indistinguishable according to the non-parametric one-tailed Mann-Whitney test). The outlier experiment with significant quality reduction when using the surrogate can be caused by a high dispersion of the surrogate predictions, or by pipelines/meta-features with an outlier training distribution.

Table 3: Comparison of AutoML approaches on OpenML datasets.

| Criteria | Non-surrogate | Proposed surrogate approach |
|---|---|---|
| ROC AUC | 0,69 | **0,692** |
| Time before best result, min | 690 | **30** |

## 6 FUTURE WORKS

Future research will focus on (1) extending the evaluation of our framework and (2) improving pipeline ranking methods.

To enhance robustness, we will expand the number of datasets, ensuring broader meta-feature representation. While our surrogate model is trained on classification tasks, assessing its performance on regression datasets is crucial for evaluating generalizability.

Further development will explore additional data modalities, such as text and images, by integrating new feature engineering strategies. LLMs could aid in dataset analysis for pipeline selection or generate auxiliary inputs to improve surrogate model predictions. Investigating their role in this context could provide valuable insights for future improvements.

## 7 CONCLUSIONS

In the paper, we propose the flexible Meta-AutoML approach that combines the fast surrogate-assisted selection of initial assumptions of modelling pipelines with variable structure.

**Analysis of obtained results** The results can be considered as a proof-of-concept for the idea of a configurable multi-stage Meta-AutoML based on surrogate estimation of pipelines using GNN. It confirms that the correct choice of the surrogate model allowed a significant reduction in the time needed to design the modelling pipelines. However, a more detailed experimental study is still needed to estimate the efficiency of the meta-approach for a large number of different ML tasks.

**Limitations** The main limitation of the proposed approach is the assumption that the similarity of embeddings for datasets is related to the similarity between modelling pipelines. However, this may be incorrect in many cases. The evolutionary stage of the proposed approach still allows the pipeline to be improved, but the convergence time can be increased for cases with an unsuccessful choice of initial assumption.

## 8 CODE AND DATA AVAILABILITY

The software implementation of the proposed algorithms and scripts for the conducted experiments are available in the open repository https://github.com/ITMO-NSS-team/GAMLET.

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

Table 4:    Meta-features used to describe datasets.

| PyMFE | | OpenML |
|---|---|---|
| Entire meta-dataset features | Averaged over meta-dataset columns features | Entire meta-dataset features |
| attr_to_inst | attr_ent | MajorityClassSize |
| class_ent | eigenvalues | MinorityClassSize |
| eq_num_attr | freq_class | NumberOfClasses |
| gravity | joint_ent | NumberOfFeatures |
| inst_to_attr | kurtosis | NumberOfInstances |
| nr_attr | max | NumberOfNumericFeatures |
| nr_class | mean | NumberOfSymbolicFeatures |
| nr_cor_attr | min | – |
| nr_inst | mut_inf | – |
| nr_num | range | – |
| ns_ratio | skewness | – |
| – | var | – |

## A    SURROGATE QUALITY ACROSS DIFFERENT PIPELINE EVALUATION METRICS

During the experiments, it was noticed that surrogates work better for pipelines evaluated using the ROC AUC metrics compared to those using the LogLoss metric. To exclude any anomalies, the average value of the relevance prediction and the standard deviation for each candidate in the set are shown in Figures 2 and 3 for pipelines evaluated using ROC AUC and LogLoss metrics, respectively. The experiment was conducted with candidate sets consisting of 10 pipelines. According to the plots, on average all the surrogates rank both types of pipelines correctly. Standard deviation of prediction is similar across all the surrogates. Overall, all the surrogates exhibit high standard deviation of prediction. Thus, predicting capabilities of models is not arguable but demanding for better training of the models.

## B    SURROGATE QUALITY ACROSS DIFFERENT SIZES OF CANDIDATE SET

In this section, we examine the behavior of the proposed surrogates on candidate sets of varying sizes. The candidate set sizes analyzed are [1, 2, 3, 4, 5, 10, 15, 20, 25, 30, 35]. Meta-datasets are picked randomly with meta-features taken from OpenML. As depicted in Figure 4, all surrogates exhibit similar Kendall-Tau correlation across these candidate set sizes. However, the precision of the surrogates decreases as the candidate set size increases. For RankNet-based surrogate this behavior is expected, since this surrogate maps any pipeline to a score. For the other surrogates we can claim that they are non-dependent on candidate pool size. Figure 5, similar to the previous section, displays the average candidate relevance (standard deviation is omitted for the sake of chart readability) to approve Kendall-Tau correlation stability over various sizes of candidate pools. According to these charts, the surrogates generally predict accurately, though the Late Fusion SetRank encounters minor issues when ranking large candidate pools.

## C    META-FEATURES

In this work we have considered meta-features to make a vector representation of datasets to be passed through surrogate model in conjunction with pipelines. We explored usage of PyMFE and OpenML packages to achieve it (Table 4).

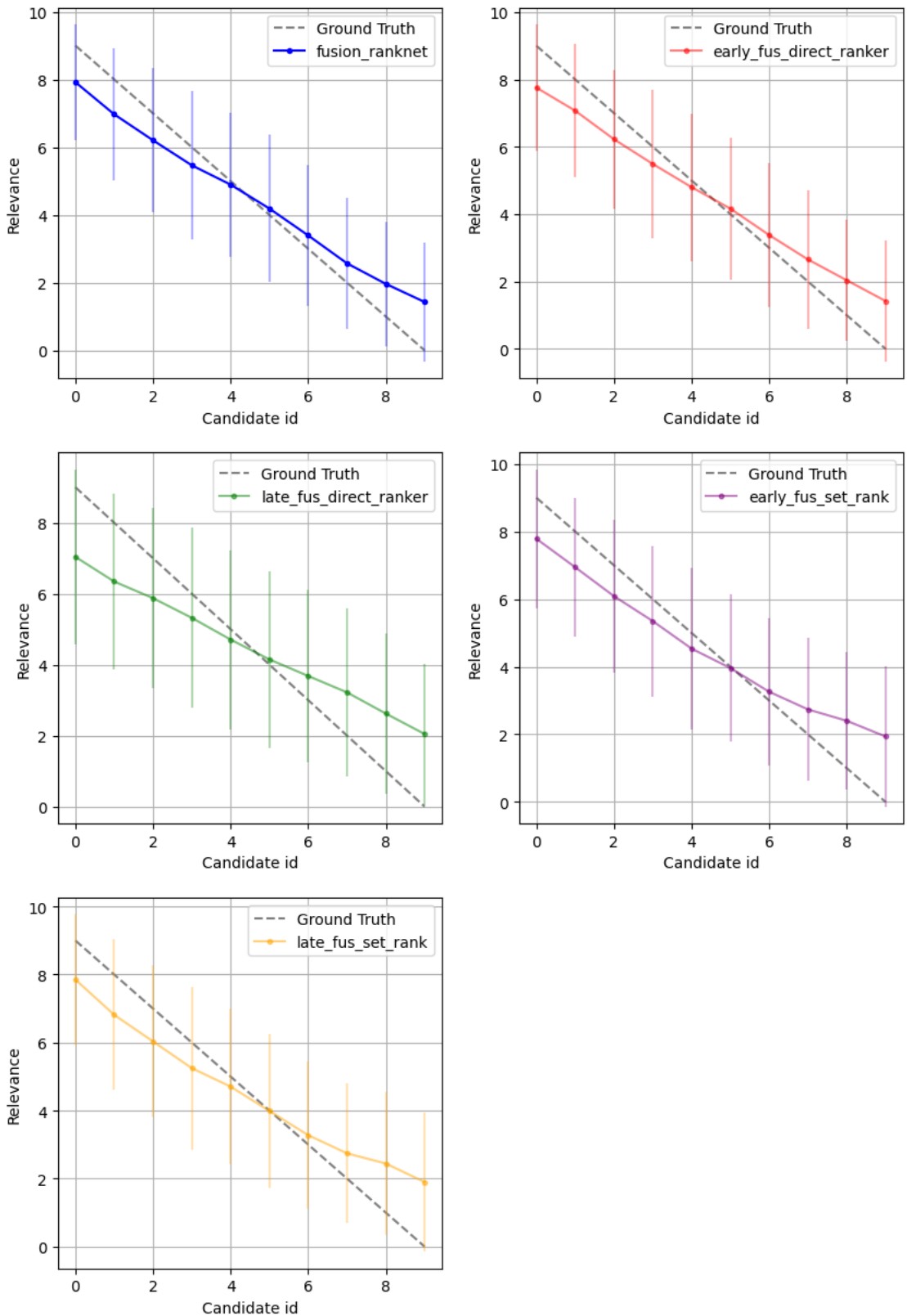

Figure 2: Predicted relevancy over true relevancy for pipelines of ROC AUC score.

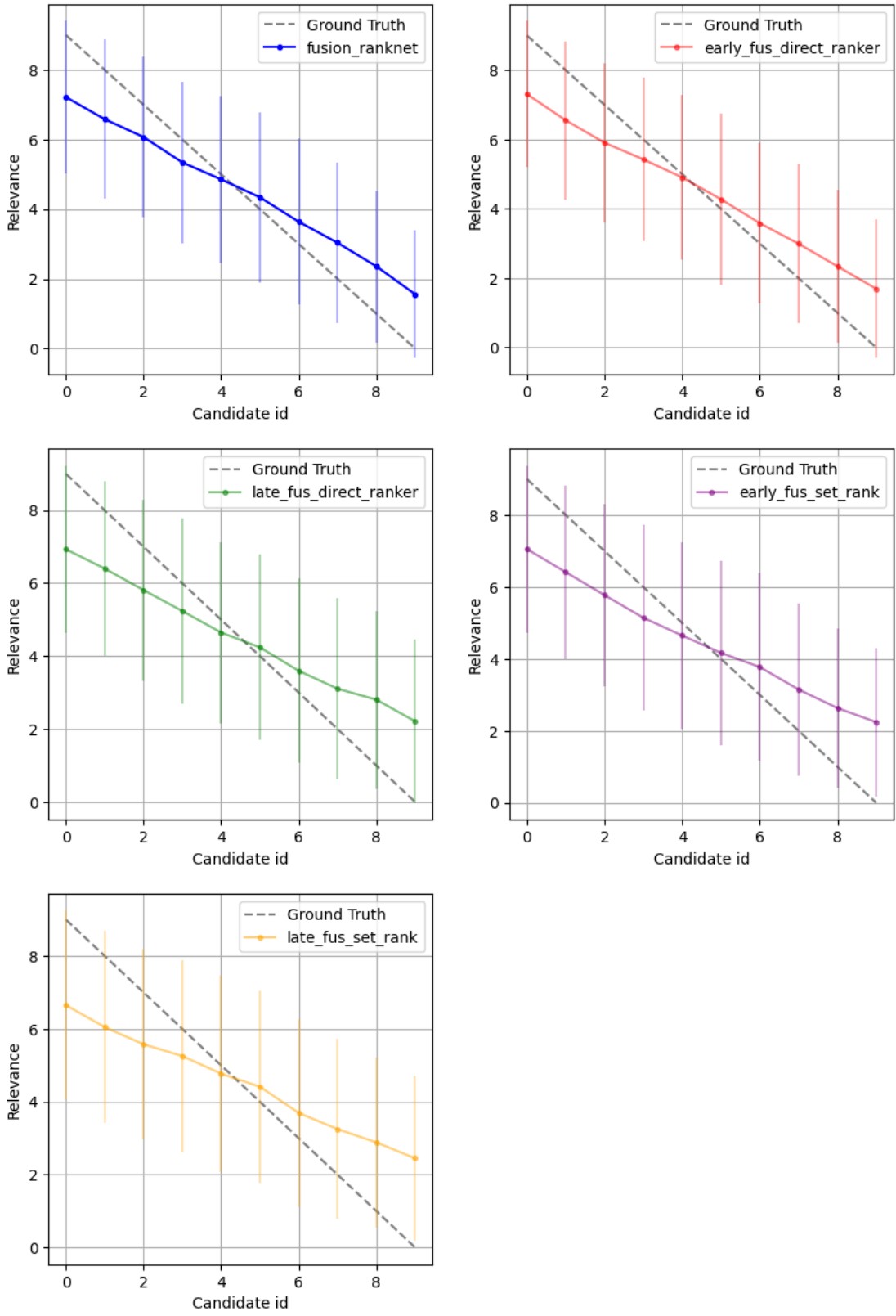

Figure 3: Predicted relevancy over true relevancy for pipelines of Logloss score.

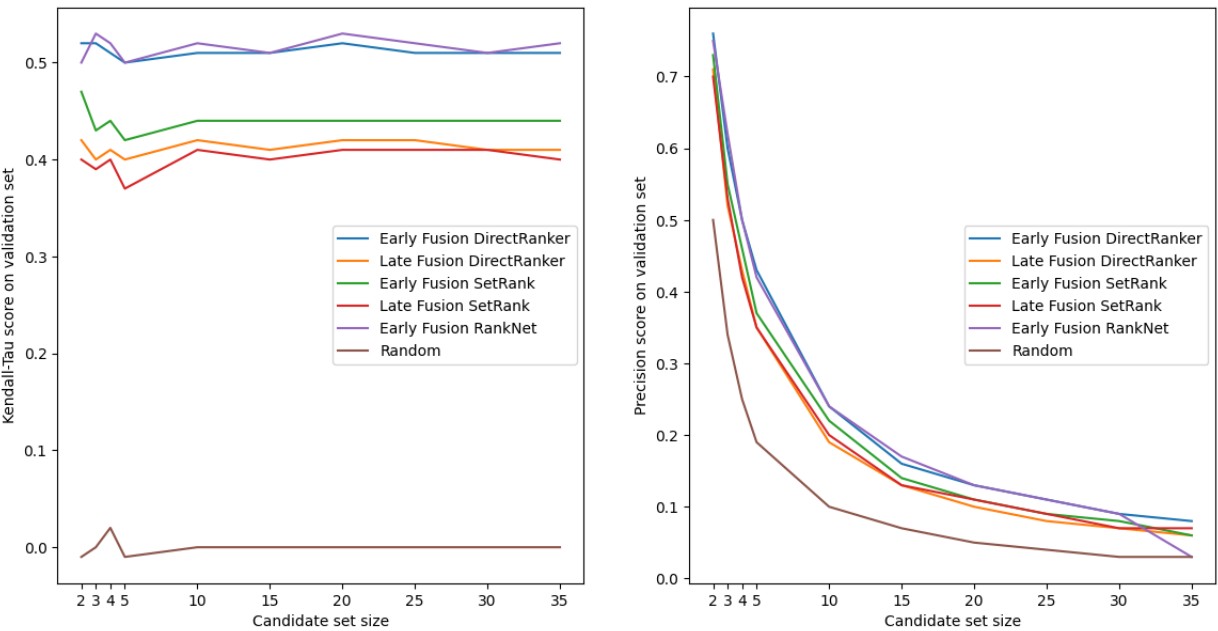

Figure 4: Surrogate quality for candidates set of different size.

Table 5: Comprehensive comparison of surrogate models.

| | AutoGluon | Our | Our w/ surrogate | Baseline | Best Baseline | Best Pipeline |
|---|---|---|---|---|---|---|
| | RocAuc | | | | | |
| kddcup09_appetency | 0,836 | 0,833 | 0,696 | 0,820 | **0,842** | – |
| guillermo | 0,915 | – | 0,882 | 0,897 | **0,917** | – |
| albert | **0,770** | 0,726 | 0,728 | 0,756 | 0,766 | 0,676 |
| christine | 0,824 | – | 0,804 | 0,817 | **0,829** | – |
| numerai28_6 | 0,522 | 0,525 | 0,519 | 0,524 | **0,532** | 0,510 |
| amazon_employee_access | 0,865 | – | 0,705 | 0,843 | **0,874** | – |
| airlines | **0,731** | 0,713 | 0,674 | 0,709 | 0,728 | 0,650 |
| sf-police-incidents | – | – | **0,678** | 0,645 | 0,676 | – |
| | Time, sec | | | | | |
| kddcup09_appetency | **537** | 15734 | 1120 | 0 | 0 | 0 |
| guillermo | **16367** | – | 32072 | 0 | 0 | 0 |
| albert | 7416 | **1334** | 1491 | 0 | 0 | 0 |
| christine | **1851** | – | 3283 | 0 | 0 | 0 |
| numerai28_6 | 5622 | 14443 | **1863** | 0 | 0 | 0 |
| amazon_employee_access | **422** | – | 2127 | 0 | 0 | 0 |
| airlines | 8058 | 16598 | **2081** | 0 | 0 | 0 |
| sf-police-incidents | – | – | **1855** | 0 | 0 | 0 |

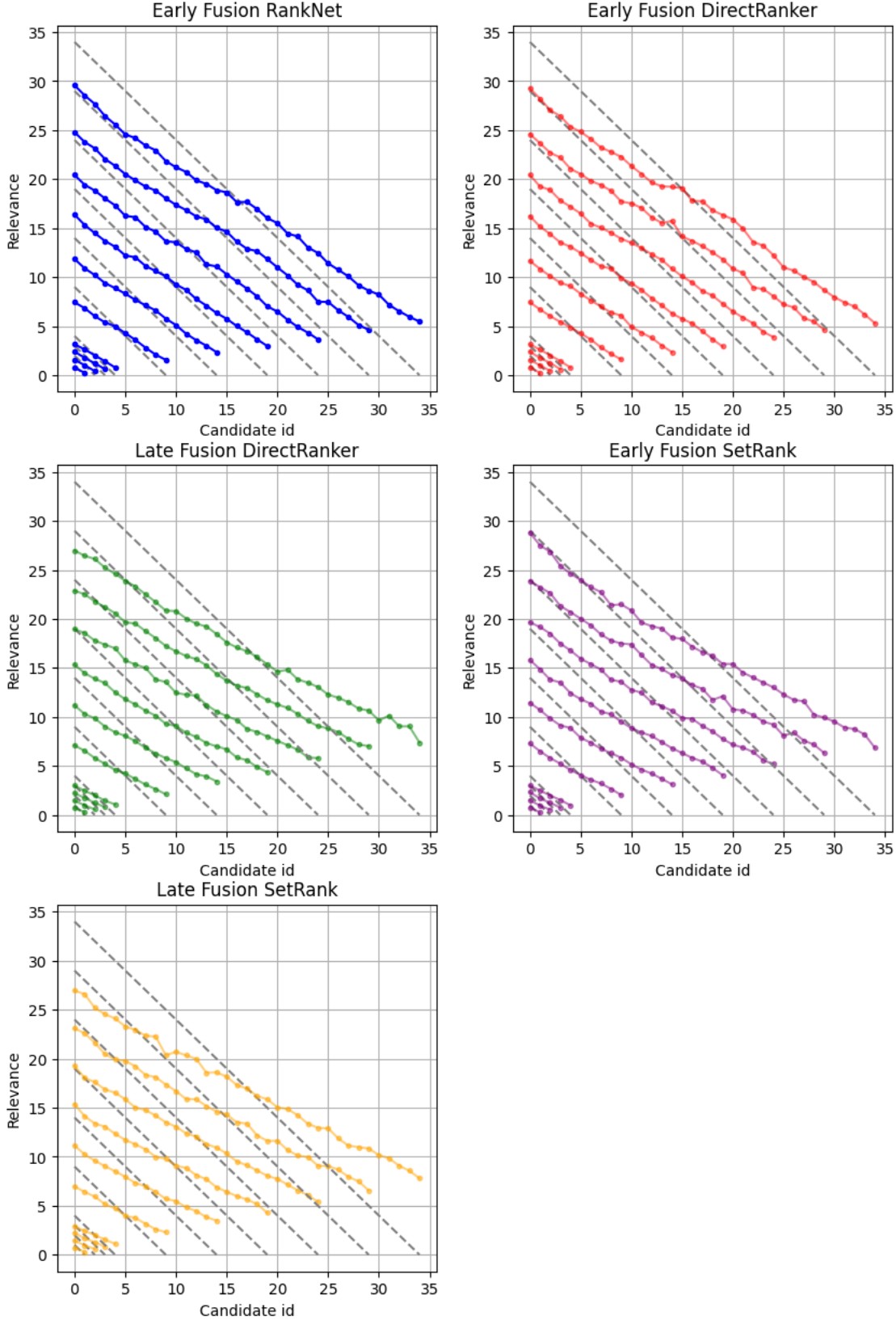

Figure 5: Surrogate quality for candidates set of different size.

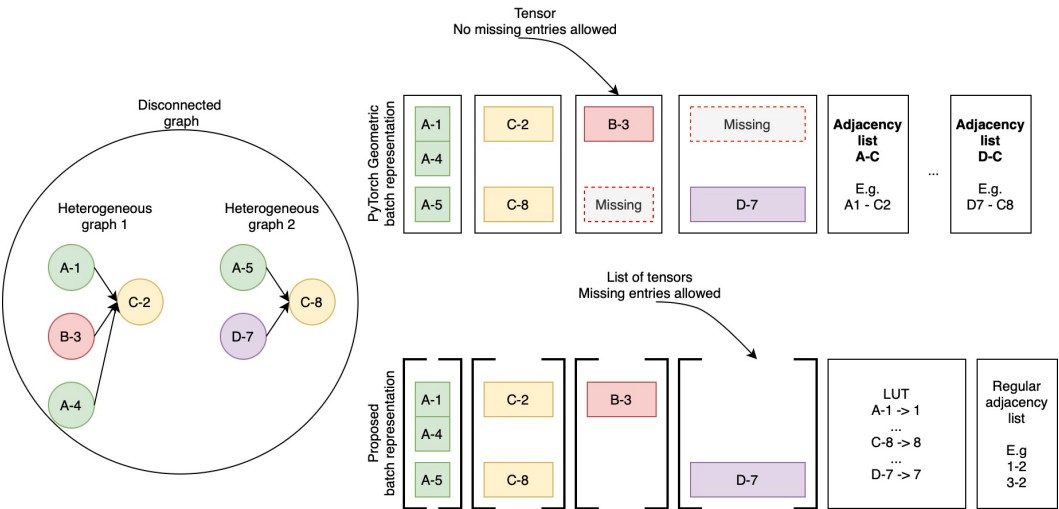

Figure 6: Batching comparison.

## D    COMPREHENSIVE COMPARISON OF SURROGATE MODELS

To assess benefits of introducing the surrogate model to our framework FEDOT we have provided experiments of held-out set of datasets. For experiments with our framework we kept all parameters same except type of optimizer (w/ or w/o surrogate). Table 5 demonstrates that using of surrogate does not always reduce time required to design a surrogate model, but it can reduce the time significantly. Currently, we have not investigated the reasons of this behavior. Final pipeline quality can vary other different datasets the pipeline is designed for. For now, we consider that the reason lies in the surrogate has a large spread of predicted values as was shown in Figures 2 and 3. In comparison with AutoGluon, our approach is not always faster (we kept all experiments with one number of jobs equal to 1), however it is worth to examine the effect of introducing the surrogate to other frameworks.

## E    HETEROGENEOUS DATA BATCHING

In this work GNN-framework PyTorch Geometric is utilized. In this framework, a batch of graphs is represented as a disconnected graph with each sample represented as a subgraph. For heterogeneous data the framework requires each subgraph to have the same set of node types. Moreover, adjacency of nodes in heterogeneous graph is considered for each pair of types separately. This format of watching is not applicable for the current work, since adjacency must be considered as it would be if all nodes be of the same type and requirement of all subgraphs to have the same set of node types limits possible combinations of pipelines in a candidate set. To overcome this limitation, a custom structure to store batch of heterogenous batch is proposed. The comparison of PyTorch Geometric batch and the proposed batch is shown on Figure 6.

In the proposed batching method each node type is stored as a list of nodes instead of tensor of nodes. This eliminates the requirement of having the same set of node types in subgraphs. A look-up table (LUT) is used to keep mapping between nodes and their indexes in the disconnected graph. This allows the use of a regular adjacency list as the nodes would be of the same type. Later, the proposed structure is parsed during graph homogenization and transformed to a regular PyTorch Geometric batch of homogeneous graphs.

## F    ARCHITECTURE OF THE FRAMEWORK

The internal implementation of each component is replaceable while preserving the external interface. The basic approach is a warm-start of optimization based on the meta-features of the dataset.

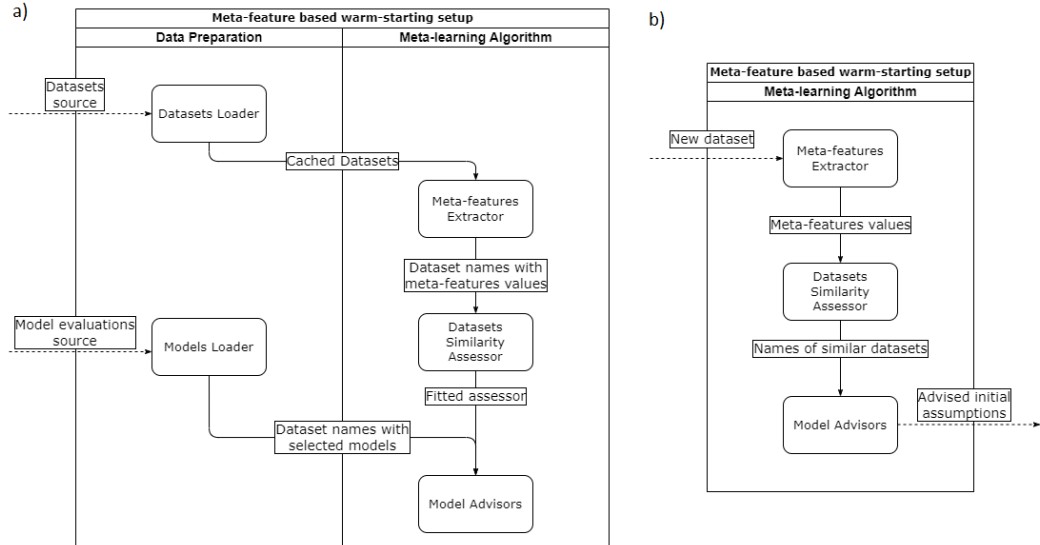

Figure 7: The interaction of the framework components (a) retrieving metadata and obtaining knowledge (b) at the stage of knowledge application

Its results can be used both independently and in any AutoML framework that starts the optimization process with some initial approximations.

The interaction between framework components at the stages of collecting metadata, obtaining knowledge and knowledge application stage is described in Figure 7.

The *Datasets loader* is designed to automate work with datasets. This component is responsible for the following operations: receiving a dataset from a dataset source, saving a dataset into a cache file, and loading a dataset into the working memory from the cache file.

The *Meta-features extractor* is designed to automate meta-features extraction. It takes as input a list of cache objects of datasets and returns a table of meta-features, in which each row corresponds to a dataset and each column to a meta-feature. Since extracting meta-features is a time-consuming process, the meta-feature values for datasets are also cached and are not recalculated when retrieving them again.

The *Datasets similarity assessor* is designed to assess the similarity of datasets by meta-features. At the knowledge acquisition stage, a table of meta-features of datasets is input. At the knowledge application stage, the component accepts meta-signs of new datasets and converts them into lists of similar datasets from the previously "memorized" ones. Optionally, it can also return a measure of similarity (or distance) between the new datasets and the "memorized" ones.

The *Models loader* component imports and unifies data from model evaluation sources on datasets. A selection of experiments from the source according to a predefined criterion is also available, e.g., selecting the best or worst models for each dataset.

*Models advisor* is the final component of the initial approximation fitting. The current implementation combines results from Models loader and Datasets similarity assessor. The first one provides a set of models for datasets, the second one provides similar datasets from "memorized" ones.

## G RANKING METHODS

For clarity, the methods proposed in Section 4 are summarized here, accompanied by schematic representations for reference."

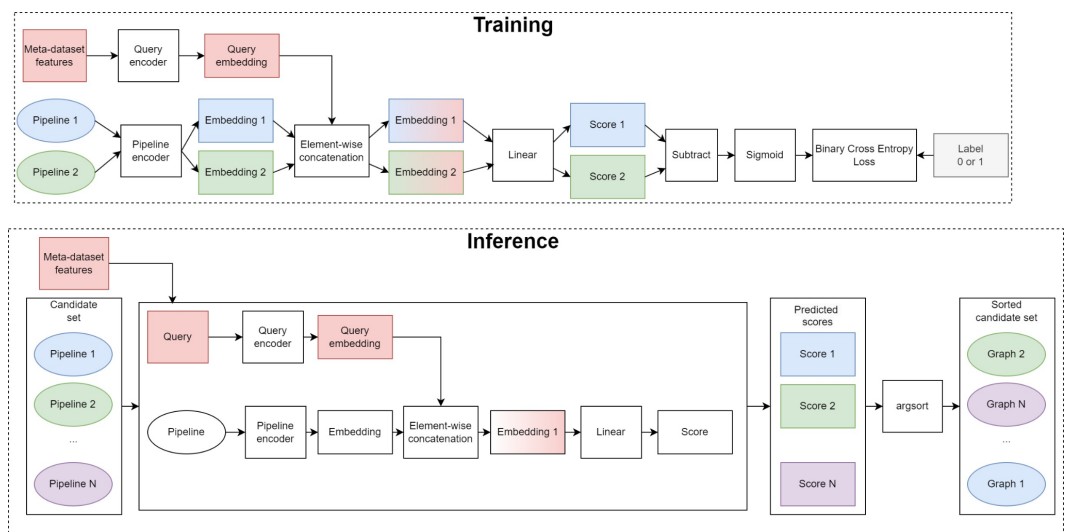

Figure 8: RankNet method scheme as outlined in Eq. 4

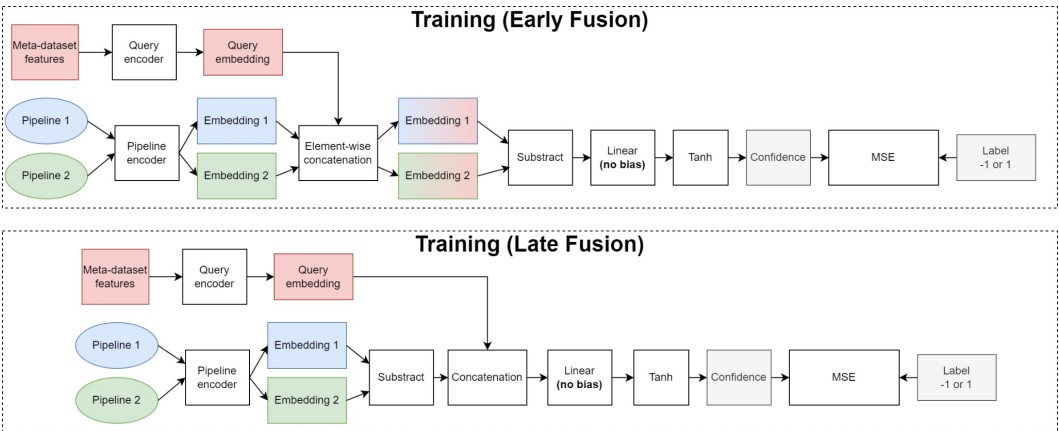

Figure 9: DirectRanker method scheme as outlined in Eq. 5 and 6

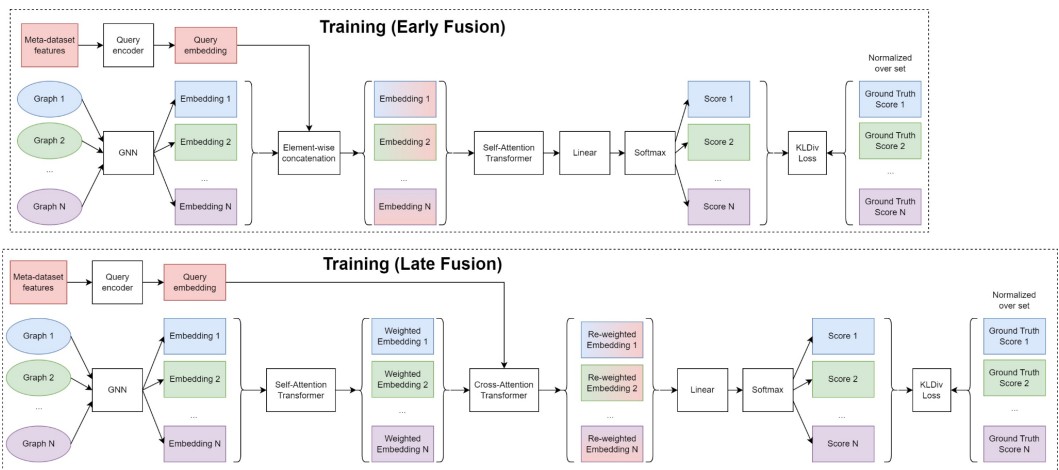

Figure 10: RankNet method scheme as outlined in Eq. 7 and 8

