# OpenReview forum: "Deep GNN-driven Surrogate for the Better Meta-learning in AutoML"
_mathai.club/MathAI/2025/Conference — MathAI 2025 Oral_

### Official Review · Reviewer_5rns · 2025-02-27
**The article presents useful practical results, an original approach, but the limitations of the proposed method are not entirely clear.**

**Rating:** 7
**Confidence:** 4

**Review:**

Strengths:

Originality: The article introduces a novel approach to Automated Machine Learning (AutoML) using Graph Neural Networks (GNNs) for surrogate modeling and ranking of ML pipelines. Combining GNNs with meta-learning and evolutionary algorithms is innovative and could significantly enhance AutoML efficiency.

Practical Impact: The proposed method reduces pipeline design time, which is crucial for large datasets and industrial applications with limited resources.

Modularity: The approach is modular and flexible, making it adaptable to various AutoML systems and tasks.

Experimental Validation: The authors compare their method with existing solutions like AutoSklearn 2.0, demonstrating reduced pipeline generation time without significant quality loss.

Weaknesses:

Reliability of Results: While the approach shows promise, its reliability across diverse data types (e.g., text, images, time series) is unclear. The article lacks a detailed analysis of its effectiveness on heterogeneous data.

Limited Dataset Scope: Experiments were conducted on only 11 classification datasets from OpenML, raising questions about scalability to broader tasks.

Limitations: it is not clear how to assess whether the method is suitable for a specific task and what kind of increase should be expected.

Conclusion:
The article presents an innovative and promising approach to AutoML, but its generalizability and reliability require further validation on a wider range of data and tasks.Despite this, Auto-ML is an extremely important area in modern big data processing pipelines, and the results shown in the work are clearly significant and worthy of attention.

---

### Official Review · Reviewer_iosa · 2025-02-27
**This paper presents a promising GNN-based meta-learning approach for AutoML that efficiently ranks pipelines to reduce computational costs, though its evaluation is somewhat limited to tabular classification tasks**

**Rating:** 7
**Confidence:** 4

**Review:**

---

### 1. **Summary**
The paper proposes a meta-learning approach for Automated Machine Learning (AutoML) that employs Graph Neural Networks (GNNs) as surrogate models to rank candidate pipelines efficiently. By leveraging a meta-knowledge database of past pipeline evaluations, the method accelerates evolutionary optimization by reducing costly pipeline evaluations. The authors implement their approach as an open-source tool, demonstrating reduced optimization time while maintaining competitive performance on tabular classification tasks. Key contributions include a GNN-based surrogate for pairwise pipeline ranking and integration with evolutionary search, validated on OpenML datasets.

---

### 2. **Strengths and Weaknesses**

#### **Originality**
- **Strengths**: Combines GNNs with evolutionary AutoML for pipeline ranking, a novel application. Extends meta-learning to graph-structured pipelines, differentiating it from prior work on simpler pipelines.
- **Weaknesses**: While the combination of GNNs and meta-learning is innovative, similarities exist to AlphaD3M (RL for pipeline search) and prior GNN-based ranking methods (e.g., https://link.springer.com/chapter/10.1007/978-3-030-88942-5_13).

#### **Quality**
- **Strengths**: Technically sound with experiments on various OpenML datasets. Ablation studies validate design choices (e.g., ranking heads). Limitations (e.g., dataset-pipeline similarity assumptions) are acknowledged.
- **Weaknesses**: Experimental details are sparse (e.g., meta-knowledge base construction, hyperparameters).

#### **Clarity**
- **Strengths**: Overall structure is logical, with clear problem formulation and methodology.
- **Weaknesses**: Dense technical sections (for example, in equations 4-8) might benefit from having more detailed "intuitive" explanations.

#### **Significance**
- **Strengths**: Reduces AutoML computation time, a critical pain point. Open-source implementation enhances practical utility.
- **Weaknesses**: Impact is limited to tabular classification; applicability to other tasks (e.g., vision/NLP) is untested. Gains of proposed solution over LLM-based AutoML (e.g. GPT-NAS mentioned in this paper) are unclear.

---

### 3. **Questions for the Authors**
1. How was the meta-knowledge base constructed? Are there biases in dataset/pipeline selection?
2. Can the surrogate generalize to pipelines with unseen algorithms or hyperparameters?

---

### 4. **Limitations**
- **Addressed**: The paper notes the assumption that dataset similarity implies pipeline similarity may fail.
- **Missing**:
  - Generalizability to non-tabular tasks and larger pipelines.
  - Computational cost of training the GNN surrogate.

**Suggestions**: Discuss compute requirements for surrogate training and scalability to diverse pipelines.

---

### 5. **Ethical Concerns**
No major ethical issues.

---

### 6. **Soundness Rating**
Claims are supported, but experiments lack depth (e.g., small dataset count, limited baselines).

---

### 7. **Presentation Rating**
The structure of the presentation is well-organized.

---

### 8. **Contribution Rating**
Novel integration of GNNs and meta-learning, but incremental over existing surrogate-based AutoML methods.

---

### 9. **Overall Score**
The approach is technically sound and addresses a relevant problem, but evaluation gaps limit impact. A broader comparison with other meta-learning-based approaches could strengthen the case.

---

### Official Review · Reviewer_Kz3W · 2025-02-27
**Semi-Supervised Meta-Learning in AutoML using a Deep GNN-Driven Surrogate Model**

**Rating:** 7
**Confidence:** 4

**Review:**

The paper "Deep GNN-Driven Surrogate for the Better Meta-Learning in AutoML" introduces an innovative approach to accelerate AutoML by leveraging meta-learning with a Graph Neural Network (GNN)-based surrogate. Instead of performing full evaluations on every candidate pipeline, the authors construct a meta-knowledge base from previous pipeline evaluations and employ a GNN to generate embeddings for complex, variable-structure pipelines. Each pipeline is represented as a heterogeneous graph where nodes correspond to machine learning algorithms—with their associated hyperparameters encoded by dedicated MLPs—and edges capture the pipeline structure. In parallel, dataset meta-features are extracted and embedded, serving as a query that guides the surrogate’s ranking of candidate pipelines.

The surrogate is trained using learning-to-rank methods (primarily a pairwise approach via RankNet) to predict which pipeline will perform better on a given task. The methodology explores both early and late fusion strategies for integrating dataset features, with early fusion yielding superior results. Experiments conducted on a large meta-dataset (over millions of pipeline evaluations) show that the surrogate achieves a Kendall Tau correlation of about 0.52–0.53—well above a random baseline—and dramatically reduces the time to reach optimal pipelines (e.g., from 690 minutes down to 30 minutes) while maintaining nearly identical predictive performance (ROC AUC).

The novelty of the work lies in combining GNN-based surrogate modeling with meta-learning to effectively handle complex pipeline structures, an advancement over prior methods that typically rely on simpler representations or reinforcement learning-based strategies. This surrogate-driven approach reuses prior knowledge to guide evolutionary search, making the AutoML process significantly more efficient.

Strengths:

- Innovative Representation: Utilizes heterogeneous graph encoding to capture intricate pipeline structures.
- Effective Ranking: Demonstrates strong ranking performance (Kendall Tau ~0.53) through a well-designed pairwise ranking strategy.
- Practical Impact: Achieves significant reduction in optimization time with minimal loss in predictive accuracy.
- Thorough Evaluation: Validated on a large-scale meta-dataset with comprehensive ablation studies confirming feature importance.

Weaknesses:

- Data Dependency: Surrogate performance may decline if new tasks exhibit characteristics not present in the training data.
- Narrow Evaluation Scope: Primarily evaluated on classification tasks; additional studies on diverse task types (e.g., regression) are needed to demonstrate broader applicability.

---

### Decision · Program_Chairs · 2025-03-08

**Decision:**

Accept (Oral)

**Comment:**

Your article has been accepted and you can give a talk on the article. All articles will be sorted by rating and within the available conference places one author from each article will be invited. If there are not enough places, then you will either have the opportunity to speak remotely or come at your own expense!